# Natural dynamics and watershed approach incorporation in urban water management: A scoping review

**Marcelo Canteiro**[1]*, **Helena Cotler**[2], **Marisa Mazari-Hiriart**[3], **Nadjeli Babinet**[4], **Manuel Maass**[1]

**1** Instituto de Investigaciones en Ecosistemas y Sustentabilidad, UNAM, Col. Ex Hacienda de San José de la Huerta, Morelia, Michoacán, México, **2** Centro de Investigación en Ciencias de Información Geoespacial, Ciudad de México, México, **3** Laboratorio Nacional de Ciencias de la Sostenibilidad, Instituto de Ecología, UNAM. Tercer Circuito Exterior Ciudad Universitaria, Coyoacán, Ciudad de México, México, **4** Posgrado en Ciencias de la Sostenibilidad, UNAM—Instituto de Investigaciones en Ecosistemas y Sustentabilidad, UNAM, Col. Ex Hacienda de San José de la Huerta, Morelia, Michoacán, México

\* mcanteiro@iies.unam.mx

**Data Availability Statement:** All relevant data are within the manuscript and its Supporting Information files.

## Abstract

Several cities are facing water emergencies related to urbanization impact and amplified by climate change. Most of the cities have responded to these crises through short-term measures. However, some cities have incorporated a watershed approach to water management in seeking more sustainable solutions. Although the importance of a watershed approach in land management is generally acknowledged, studies on this topic have typically focused on theoretical models, water management in rural areas or single case-studies of cities or countries. In this research, a scoping review of the literature was performed, based on the PRISMA 2020 statement, in three databases: Web of Science, Google Scholar and SciELO. Forty-one studies were identified analyzing 17 city cases implementing urban actions from a watershed approach in water management. These cities were from the Global North and Asian rising world powers. The lack of results of cities from the Global South, based on the research undertaken, was the main limitation and bias identified. Most of the Global South results identified in this research were theoretical models, scenarios and cases of rural areas instead of urban contexts. The results obtained indicate that the main motivations for cities to implement a watershed approach were water scarcity, floods and contamination of water bodies. The implemented actions focused on the shift from gray to green and blue infrastructure and on conservation measures. Lastly, the challenges to introduce those actions were mainly the lack of economic investment, insufficient experience, stakeholder opposition, and regulatory obstacles. Urban water management could be seen as an opportunity to change the way we relate to urban territory. Incorporating a watershed approach into urban planning and water management could promote more sustainable cities.

**Funding:** Marcelo Canteiro has a postdoctoral scholarship of the "Programa de Becas Posdoctorales de la Dirección General de Asuntos del Personal Académico" of the UNAM from September 2022 to August 2024 with number CJIC/CTIC/1167/2022. Nadjeli Babinet has a PhD scholarship from the Consejo Nacional de Humanidades, Ciencia y Tecnología (CONAHCyT), from August 2023 with CVU number 731000.

**Competing interests:** The authors have declared that no competing interests exist.

## 1. Introduction

Historically, urban centers have had a challenging relationship with water. This is due to their high water demand, related to urban population density and its commercial and industrial activities, and also due to the limited availability of this resource. In addition, cities are affected by several water problems, such as floods, periods of water shortage, and public health issues related to water quality alteration [1–4]. On the other hand, the main promoters of land-use change, habitat loss and biodiversity impact are related to urbanization processes. They also affect water dynamics, altering the natural flow of rivers, water quality, and associated ecosystems, both within the urban area and the surrounding region [5–7].

Regarding the urbanization processes impacts within urban areas, a central alteration generated is the increase of potential contaminant sources related to surface water and groundwater [8, 9]. These risks include urban water being contaminated by microbiological agents, inorganic and organic compounds [10]. The presence and increase of potential contaminant sources, such as the diversification of pollutants (i.e., microplastics and organic compounds) represent a risk for guaranteeing the availability of clean water in cities [11, 12]. In addition, land-use change related to urbanization generates a waterproofing effect on urban surfaces that prevents infiltration or percolation. This hinders or interferes with the potential for aquifers' recharge and for adequate water quality, decreasing the availability of groundwater and promoting the presence of floods [13].

Beyond the urban area, urbanization promotes water-related impacts at the regional level. Urbanization impacts are reflected in the degradation of natural areas at the headwaters of watersheds, in saline intrusion, and in the alteration of water quality in the water sources, which in several cases are outside the city borders [14, 15]. For instance, at the regional level, urban growth and expansion foster territorial and land-use modifications that translate into changes in the structure and functioning of the ecosystems that support the provision of adequate quality water resources [13]. Moreover, urban centers can modify regional water dynamics by extracting water for urban or industrial supply. In this regard, the increase of water consumption in cities is not only due to direct consumption of water as a resource. It is also related to water leaks in aging infrastructure [13], and to indirect consumption, associated with the demand of products that use large amounts of water for their operation or production, such as for agriculture and livestock, as well as other productive activities [16]. Urbanization processes are also related to long-distance impact (telecoupling) because of the urban demand for water for the production of commodities (food, energy, mining, etc.), which is fulfilled with water sources outside of the urban territory [17, 18]. Therefore, urban water demand generates regional impact at the water extraction location, where water availability decreases, and, in some cases, the quality of water sources is altered. Urban water demand can also affect water dynamics at the water extraction site or during its transportation. In addition, cities possess a regional effect by moving poor quality water to outside the city to solve sanitation and flooding problems [19]. The need to transport water to and from cities implies a high energy expenditure, and this energy is generally produced outside the urban area with the associated environmental impact. Added to all this impact is the fact that cities around the planet currently invest close to USD $5.4 billion a year in water management. This is due to the decrease or loss of ecosystem services (*i.e.* provision, regulation) related to water and caused by the actual degradation of the watersheds [2]. Lastly, the growth and densification of urban population increases water demand and competition for access to and the use of this resource between urban centers and the productive activities in the surroundings, that include different water users [20]. This might cause, in some cases, that obtaining water could lead to intensive exploitation of the supply sources which would affect the dynamics and natural

processes related to water, as well as compromise the long-term sustainability of regional water sources [21].

Urban water problems and their associated regional impact become increasingly pressing since approximately 50% of the population lives in urban centers, and by 2050 this percentage is expected to reach 70% [22]. In addition, urban problems related to water are intensified in the current context of climate change and its uncertainty. In climate change scenarios, urban areas are likely to face significant challenges, such as heat waves, floods, droughts, problems in food production and limitations in obtaining freshwater [14, 23].

## 1.1 How is urban water managed?

Urban water management usually considers water as a resource. For instance, water management generally focuses on obtaining clean water, managing stormwater and wastewater, and performing flood control, based on technological solutions that do not consider the hydrological dynamics of the watershed as a system. This understanding of water as a resource, rather than a fundamental component of a system, promotes the occurrence of socioenvironmental impact. This impact is expressed in environmental degradation caused by changes in land use that do not consider watershed dynamics. These changes are driven by the need to obtain water from outside the urban territory, through artificial transfers of water from external sources, as well as the need to transport poor quality water out of the city [24]. For example, Mexico City Metropolitan Area imports clean water from the upper part of the Balsas River watershed and exports poor quality water to the Panuco River watershed [25–27]. This kind of urban water management fosters problems of inequality, marginalization, and poverty. These processes of water transportation and cleaning, based on the use of fossil energy, have an additional operating cost. Therefore, the decision where water is distributed and how much water is disinfected or treated, as well as how these processes are performed, becomes a sociopolitical issue related to water governance issues [28]. Simultaneously, the use of technology to deal with water management, by supplanting or complementing natural processes lost or reduced by urbanization, is not always effective and efficient. Furthermore, in some cases, constructed infrastructure is abandoned with the consequent environmental impact and waste of space, time, financial resources, energy, and materials that will become obsolete [13].

Therefore, the urban water management decision-makers currently face a combination of problems: an increasing water demand related to population growth and industrialization, the deterioration or obsolescence of hydraulic infrastructure, the interests of different societal sectors and the competition for water use, as well as alterations and socioenvironmental challenges intensified by climate change [29]. According to projections of urban growth, these effects will continue to increase over time [22]. To face this scenario, it is necessary to understand how to improve water management in cities, and an adequate vision from a practical standpoint should be based on the functional hydrological units: the watersheds.

## 1.2 Urban areas inside the watershed dynamic

To incorporate natural dynamics and processes in urban water management an integral view of the territory, beyond the urban limits, is necessary. In this sense, the territory delimited and covered by a watershed is considered the optimal area to establish such management [30–32]. This is because within this territory, depending on the biophysical characteristics of the watershed, hydrological processes, such as infiltration, aquifer recharge, spring formation, connectivity between water bodies and aquatic ecosystems, will occur. A watershed encompasses the entire geographic space in which water drains towards the same point and has clear limits established by natural elements, making possible to isolate and to understand the inflow and

outflow of water from the system [33, 34]. In the space delimited by the watersheds, many of the processes that control the dynamics of the ecosystem occur, including the dynamics and water processes related to the cities immersed in the watersheds. Therefore, watersheds' functionality and integrity make them strategic ecosystem management units [12, 30, 33, 35].

Additionally, groundwater dynamics must be considered since cities are generally supplied, at least partially, by groundwater. Groundwater dynamics occur in a system with a complex structure where aquifers coexist at different depths and are characterized by belonging to water flows at different spatial and temporal scales, as well as local to regional flows that keep superficially separated systems interconnected [36]. These dynamic properties of groundwater are determined by the geological characteristics of the area. When managing the land and water sources, it is necessary to understand that surface and groundwater dynamics are different but closely related [37, 38].

Water management with a watershed approach proposes managing a geographic space, considering the watershed as a complex system. In this sense, it is based on understanding water as a central and integrative element of natural systems proposing its management through its consideration as a common good [39] instead of a commodity. Considering the watershed as a resource system and water as the resource unit requires organizing the actors involved for its governance so that everyone can benefit from the system of resources without any actors evading their responsibilities or acting opportunistically [39]. Therefore, a comprehension of the interactions that occur inside the watershed system is needed, including: those between the biophysical components determined by environmental conditions; the types of territories' appropriation regarding social organization, economy, technology; and the institutions related to water management at diverse scales and levels [31, 40, 41].

To consider the dynamics and the interactions inside the watershed could contribute to the solution of various urban water-related problems. However, although the benefits of a watershed approach are well known, the application of this approach has been difficult. The challenges for implementing this approach are related, for instance, with the requirements for respecting the natural boundaries and dynamics, which do not generally coincide with administrative limits and institutional coordination capacities at the basin level. This approach also presents a time scale that is often not compatible with the need for an urgent response to critical short-term situations and with the brevity of the political-electoral cycles [42]. Furthermore, the implementation of this approach might require an economic investment for generating urban water management changes which could be incompatible with the economic resources and investment priorities of those responsible for water management. These difficulties for the implementation of a watershed approach in space management increase when entering urban areas. The management of the territory in urban areas seems to be conceptualized as isolated from natural processes, so generally the management of urban territory is thought and executed through artificial processes mediated by technology and energy expenditure.

In this context, many cities have experienced water crises in the last decades, and some are on the way to reaching a critical situation regarding water called "Day Zero". This concept was established in 2017 by officials in Cape Town, South Africa, and marks the moment when an open water supply ends and access to water begins to be rationed, so the population must collect their portion of water every day [29]. There are many cities that will possibly reach a similar situation if the authorities do not modify their urban water management. For example, the cities of Sao Paulo, Brazil and Monterrey, Mexico experienced a dry season in 2015 and 2022, respectively. This caused water shortages, with some parts of the city only receiving water two days a week [20, 43]. In Mexico City, the capital of Mexico, part of its population experiences water scarcity, either with no water or only at certain times of the day or week [27]. In

addition, Sao Paulo in Brazil, Mexico City and Buenos Aires in Argentina, experience flooding problems related to deficiencies in water management [10, 13]. In contrast, some cities in different regions, such as Melbourne, Australia, Portland and New York in the United States, and Chinese cities such as Shanghai, Wuhan and Shenzhen, have introduced changes in urban water management to include the consideration of natural dynamics and processes related to water. These changes were achieved through watershed-based programs, as well as laws, policy modifications and specific urban interventions, to generate changes promoting long-term urban sustainability [44–46]. At a country level, New Zealand incorporated since 1868 a watershed approach in national water regulations and implemented a territorial organization around river basins [47].

Even if there are published case studies and literature reviews on the importance of a watershed approach in land and water management, they generally study cases in rural or natural areas, are single case-studies of cities or countries, or compare conceptual methods or models related with the watershed approach instead of comparing cases with an actual implementation of this approach [48–51]. Thus, in order to learn from the experience of cities worldwide which implemented changes in urban water management to include natural dynamics and a watershed perspective, a broader understanding is needed. Specifically, it is necessary to assess which are the drivers for cities to implement those actions, the kind of actions implemented from this approach in urban areas in diverse regions, as well as the main challenges and results obtained. Through a scoping review, this research aims to analyze how paths towards urban sustainability could be generated by incorporating the consideration of the dynamics and natural processes related to water, in urban water management, through a watershed approach. This research was performed through a scoping literature review [52] and guided by the following research question: What is known from the literature about the incorporation of a watershed approach in urban water management? This study aims to identify the existing research related to the incorporation of natural dynamics and process into urban water management, as well as to recognize any existing gaps in knowledge.

## 2. Methodology

The methodology followed the PRISMA statement through the "PRISMA-ScR checklist" (S1 Table in S1 File) [53], which allowed implementing a robust, clear, and transparent method for reviewing the literature [53, 54]. The review focused on cases of cities that have implemented changes in their water management, through actions in the urban area, to include a watershed approach and address water challenges. These cities were identified and analyzed using an explicit and reproducible method in the search, evaluation, and synthesis of information [52]. This scoping review allowed us to answer the research question through rigorous planning and strict eligibility criteria for the cities included as case studies [27]. The steps performed in the literature review process are explained below. All the research process, including the search of information in databases, the implementation of exclusion and inclusion criteria, and the data collection and analysis, was implemented in the period between September 2023 and January 2024. The most recent search was implemented in January 2024 (Step 6).

### Step 1: Research questions

The initial research question for the investigation was: What is known from the literature about integrating the watershed approach into urban water management? This question was essential for guiding the scoping review and defining the limits of the research. Additionally, four specific questions were formulated: 1) What are the main drivers reported in the literature for incorporating a watershed approach into urban water management? 2) What are the main

actions identified in the literature to integrate a watershed approach to water management in cities? 3) What are the main challenges identified in the literature for cities to integrate a watershed approach in their water management actions? 4) What are the main results reported in the literature related to the implementation of the identified actions?

## Step 2: Search strategy

In order to identify suitable case studies for analysis, a search was conducted to identify cities that had incorporated, to some degree, a watershed approach in their water management. Such a search was performed in three databases: Web of Science, Google Scholar and SciELO (Scientific Electronic Library Online) and included scientific works that partially or generally included a watershed approach in urban water management. The search for information used the following combination of key words: "watershed" and "urban water management"; "watershed" and "cities water management"; "river basin" and "urban water management"; "watershed" and "policy" and "urban" and "water management"; "watershed management" and "urban" and "case" and "water management"; "watershed approach" and "urban" and "case" and "water management"; "water sensitive" and "watershed" and "cities" and "water management"; "water resources management" and "urban" and "watershed" and "case"; "socioecosystem" and "urban water management" and "case"; and "socio-ecological system" and "urban" and "water management" and "case."

## Step 3: Selection criteria and eligibility of documents

The initial phase of the document review process involved in the examination of the first 20 search results in each database for each keyword combination. All the documents identified in the databases that were not part of the first 20 results were excluded. Within those first 20 results, inclusion and exclusion criteria were applied to evaluate and screen the documents [54]. Related to the applied criteria, the first filter was that the documents had to be written in Spanish or English. All identified documents that were in a language other than Spanish or English were excluded. Afterwards, the documents' title and summary were analyzed in order to identify if the document content was related to the research question. Only documents that could be accessed based on the institutional access of the Universidad Nacional Autónoma de México (UNAM in Spanish), that presented a title and a summary that allowed knowing the scope and objectives of the investigation and that were related with the research question were selected [27]. The inclusion criteria focused on documents that provided information about urban water management with a watershed approach or that included natural dynamics of water in water management in a specific city or a case comparison between two or more cities. The exclusion criteria, conversely, removed any documents that focused on general water management, water management in rural areas, or water management at a national, regional, or global level.

## Step 4: Data collection

All the documents found were analyzed to determine if they contained the information sought through the establishment of mandatory and complementary categories of information. These categories were divided in two compulsory categories and two complementary categories. The compulsory categories were as follows: policies or actions that include a watershed approach (Actions) and problems solved through these policies or actions (Drivers). In addition, the two complementary categories were the following: positive consequences and negative consequences (Results), as well as the challenges or barriers to implementing these policies or actions (Challenges).

**Table 1. Cities selected as case studies and references from the scoping review.**

| Continent | Country | City | Reference |
|---|---|---|---|
| Asia | China | Tianjin, Tianjin | [45, 55] |
| | | Shanghai, Shanghai | [56–58] |
| | | Chongqing, Chongqing | [56, 57, 59, 60] |
| | | Wuhan, Hubei | [56, 57, 59, 61, 62] |
| | | Nanning, Guangxi | [56, 57] |
| | | Ningbo, Zhejiang | [56, 57,59, 63, 64] |
| | | Shenzhen, Guangdong | [56, 57, 59, 65–67] |
| | South Korea | Seoul, Seoul | [46, 68, 69] |
| | Singapore | Singapore, Singapore | [44, 45, 57, 65, 70–75] |
| Oceania | Australia | Melbourne, Victoria | [44, 45, 76–80] |
| Europe | Germany | Berlin, Berlin | [45, 81] |
| | Netherlands | Rotterdam, South Holland | [44, 63, 82–85] |
| | Austria | Vienna, Vienna | [86–88] |
| | Sweden | Stockholm, Stockholm | [89] |
| North America | United States of America | Portland, Oregon | [44, 63, 90] |
| | | Philadelphia, Pennsylvania | [45, 91] |
| | | New York, New York | [46, 92] |

Based on this analysis, the last filtering of documents was performed, leaving only the documents containing information of at least one of the mandatory categories, and the data were recorded. In the cities where information from the mandatory categories was identified; the complementary categories were registered. Documents that analyzed cases with theoretical models, recommendations, or scenarios for watershed implementation, which failed to include specific actions or policies implemented, were excluded. Only documents with specific actions or policies implemented for solving urban water problems within specific cities were included in the following step of analysis (Table 1).

Two reviewers were in charge of applying the selection criteria and collecting the data from each document. First, the reviewers worked independently in the literature review process and then comparing and double-checking the work collected by their peer during this process to confirm the data included was accurate.

## Step 5: Data analysis

The collected data were analyzed to understand what changes were undertaken in each city to incorporate natural dynamics in water management from a watershed approach. For each document, the cities analyzed were identified and each document was classified according to the city or cities addressed in each study (Table 1). Subsequently, all pertinent information corresponding to the mandatory categories (drivers and actions) and the supplementary categories (challenges and outcomes) were identified for each city, based on an exhaustive review of all pertinent documents. That information was classified for each category (actions, drivers, results and challenges) and analyzed in the content and the Tables 2–5, details in the Results section. Lastly, the data analysis allowed identifying the implications of these incorporations in the transition towards more sustainable urban conditions and the challenges of implementing these water management initiatives for cities, details in the Discussion section.

## Step 6: Complementary data collection and analysis

Based on the data analyzed (Step 5), some information gaps were identified in the cities analyzed regarding the drivers, actions, challenges, or results for incorporating watershed

Table 2. **Drivers for including a watershed vision in cities water management based on the selected case studies.**

| DRIVERS | Number of cities presenting this driver | Percentage of the total of cities analyzed which present this driver |
|---|---|---|
| **Lack of water supply** (drought, intensive exploitation, or low water quality) | 15 | 88 |
| **Flood risk** | 15 | 88 |
| **Surface water contamination** (urban runoff, nutrients, blooms, or health issues) | 13 | 76 |
| **National policy drivers** (laws, budgets, or programs) and political will (legislative and executive powers) | 13 | 76 |
| **Improve habitability** and recreational spaces in cities | 12 | 71 |
| **Mitigate urbanization effects** on the environment | 8 | 47 |
| **Economic incentives** (high economic costs of traditional water management, or potential economic benefits from changes in water management) | 5 | 29 |
| **Climate change and uncertainty** (higher resilience motivation or direct experience of climatic extremes events) | 4 | 24 |
| **Social demands** (citizens' demand for environmentally friendly policies or collaboration between activists, academics, private sector, and policy makers), **and historical drivers** (historical and geopolitical drivers for local water sources) | 4 | 24 |

dynamics in water management. For instance, the analysis of the drivers, actions, challenges, and results revealed, in some cities, a potential for bias due to the limited number of documents identified. This was addressed through the implementation of an additional step in the methodological process. For this step, a search was undertaken specifically for all the cities identified (Step 4) and analyzed (Step 5). Such a search was performed in the same three databases (Step 2). The search for this complementary information used the following combination of key words: "city name" +" urban water management". Afterwards, the same selection criteria were applied (Step 3) and the list of documents regarding each city was updated with the new results (Table 1). Lastly, the complementary information was analyzed to have a more robust understanding of the main actions, challenges, drivers, and results for each city (case study) incorporating a watershed approach in water management (Step 5), details in the Results and the Discussion section.

## 3. Results

The literature review based on the PRISMA 2020 statement proposed methodology (Step 1 to 5) yielded 121,628 items. A total of 359 documents were subjected to review, with 94 ultimately selected based on the implementation of the inclusion and exclusion criteria (Step 3). The 94 documents selected were analyzed in-depth and filtered again based on the defined criteria and the fulfillment of at least one of the mandatory categories of information required (Step 4), leaving 9 documents that fulfilled all the criteria and included the required information (Fig 1). From these 9 documents, 17 case studies were selected to be analyzed in this research (Table 1). Based on the Step 6, 32 additional documents were included to strengthen the analysis of these 17 case studies. The information details and content of the 41 documents included in this scoping review can be found in the supplementary information (S2 Table in S2 File). Additionally, the list of the 94 documents selected that were analyzed in-depth is included as a supplementary information (S3 Table in S3 File).

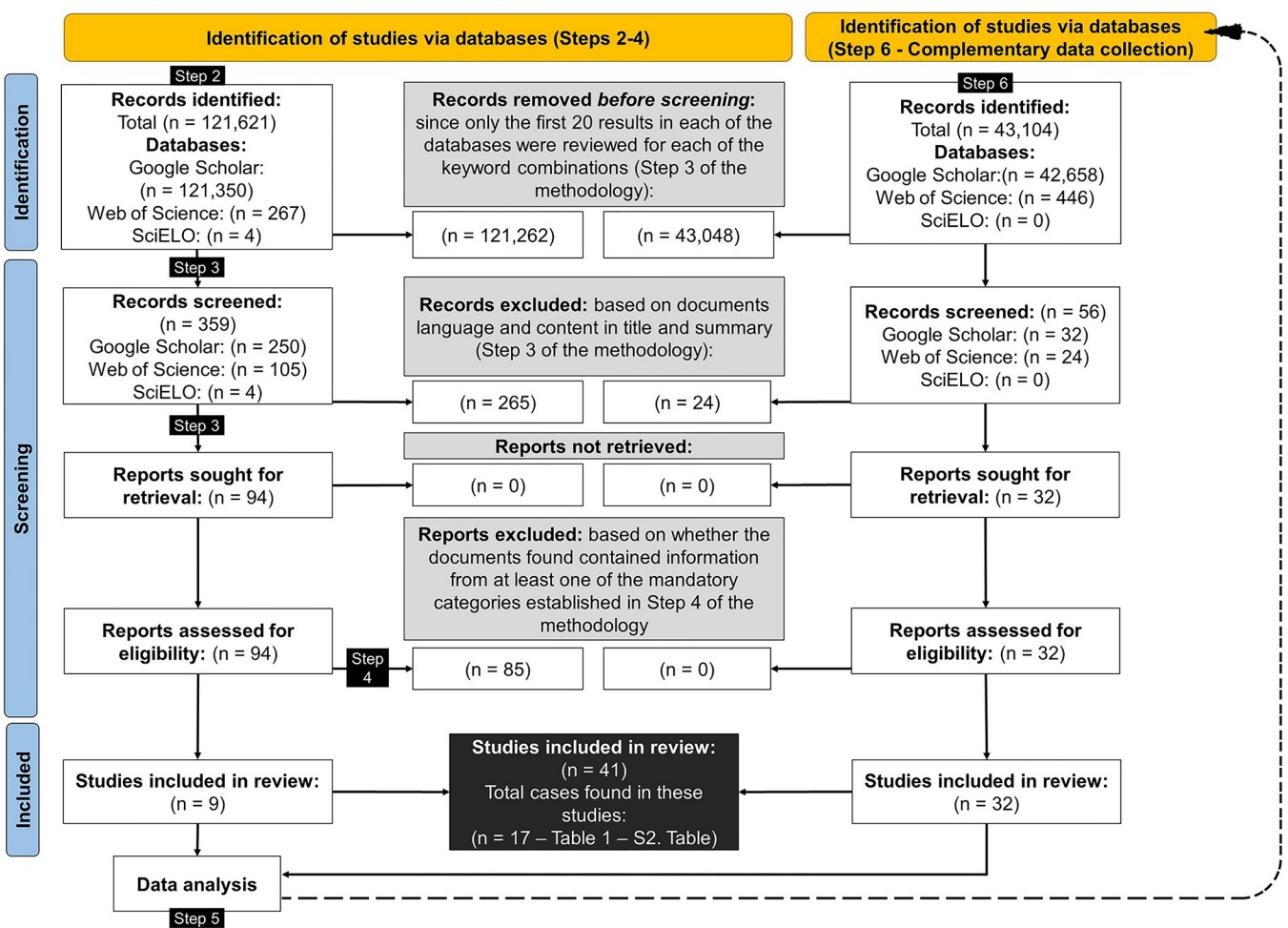

**Fig 1. Diagram of the literature search and screening process based on PRISMA flow diagram [54].** Details of the inclusion and exclusion criteria are shown in the methodology Steps noted in black in the figure.

Based on the described results, it can be observed that all the documents identified are from the 21st century, and the majority are from the last 10 years. This shows that the incorporation of a watershed approach in urban water management is an innovative issue and that few cities have been able to integrate measures in this context to address problems related to urban water. It is also important to note that the majority of the cities identified correspond to cities in the Global North (Europe, North America, and Australia) and three emerging global powers (Singapore, South Korea, and China). The latter have experienced significant economic growth in recent years [93, 94]. Nevertheless, some studies originating from cities in other countries belonging to the Global North, such as Japan and Canada, were excluded from the review due to methodological criteria. This was because the documents found were conceptual studies, case studies focusing on a country or regional scenario instead of an urban case, or because the documents found had characteristics that did not fulfill the inclusion criteria of this review [95–99]. It is interesting to note that the studies found from cities of countries of the Global South (such as India, Mongolia, Ethiopia, Ecuador and Bolivia) did not fulfill the inclusion criteria posed by the methodology used in this review. These studies were excluded mainly because they were cases in rural contexts, not urban areas, conceptual studies or theoretical modeling of scenarios instead of case studies related to an actual implementation of

actions and policies in urban areas [100–105]. The absence of cases from the Global South is of significance when considering the fact that, for instance, Latin America is characterized by a high degree of urbanization, with urbanization projections for this region also being high for the coming decades [106].

The experiences of the cities analyzed in this scoping review were synthesized through the identification of the following: a) drivers, b) actions, c) challenges faced by cities and d) results obtained by including a watershed approach in urban water management. Those results have been organized in Tables 2 to 5 which show the main categories of drivers, actions, challenges, and results identified based on the cities analyzed. For each topic, the tables indicate the number of cities that fall under each category (main drivers, actions, challenges and results), as well as the percentage that this number represents over the total cases of cities analyzed.

## 3.1 Drivers of change

The main drivers that motivated changes in urban water management were associated with urban water problems, such as water scarcity and floods. Water availability for satisfying urban demand was a motivation in 88% of the cities analyzed (Table 2). This motivation was mainly related to the presence of prolonged droughts, low quality of water sources or the intensive exploitation of water sources in those cities, which limits water availability and promotes water scarcity. Besides, 88% of the cities analyzed were forced to change their water management policies because of flood problems (Table 2). In order to address flood problems in Melbourne, Portland and Berlin, those cities incorporated measures such as the rehabilitation of surface water sources and associated ecosystems generating corridors and preventing surface runoff [44, 45, 56, 63, 79].

Besides, it was observed that environmental and social drivers, related to urban surface water quality, motivated changes in urban water management. For instance, 76% of the cities analyzed included water management changes to face water quality alteration problems in their surface water bodies, as well as aquifers (Table 2). Concern for water quality is mainly based on water availability, issues of public health, the recreational use of natural spaces and the habitability of the city. In this sense, for Chinese cities, even in cities where water resources are abundant, poor water quality is a threat for water security [60, 67]. In order to respond to this problem, through the "Sponge City" program, China promoted the recycling of rainwater and wastewater treatment to optimize the urban water drainage system for water distribution and wastewater purification [56]. While in Philadelphia, in the United States, actions were performed to reduce runoff, to decrease contaminants input, and to prevent the overflow of the combined sewage system, as well as to address health issues such as yellow fever outbreaks [45, 91].

Regarding political or institutional motivations for changing urban water management, some cities have made changes in their water management because of regional and national policies, regulations and budgets that promoted a watershed approach incorporation (76% of the cities) (Table 2). The majority of these cities are located in Asian countries, such as China and Singapore, where a robust institutional framework guides the implementation of necessary changes. This framework is characterized by a top-down approach, which enables the effective and efficient implementation of required changes [56, 57, 70–74]. For instance, in Chinese cases, the cities were used as pilots to test new nationally developed water management strategies and frameworks, known as the "Sponge City" concept [56, 57], which is a framework developed to address and overcome water problems related to surface runoff, water quality, water storage, and climate change through green infrastructure applications, for example, green roofs, rain gardens or bioretention [57]. China released the national strategy for the

Sponge City construction through programs, policies, incentives, and the selection of some cities to act as pilots in the application of this water management framework [56, 59]. In other cities, such as New York, Melbourne, Seoul and Berlin, the existence of consolidated institutions with national frameworks, regulations and budgets promoting water management changes were identified as key drivers for implementing actions [76, 79, 81, 92].

Habitability was reported in 71% of the cities as a driver for introducing changes in urban water management (Table 2). This was measured by a range of life quality factors, including access to fresh water and the possibilities of performing recreational activities in the city. For instance, in Melbourne and Vienna, some modifications to urban water management were prompted by the impact of current management practices on recreational activities due to alterations in the water quality of some recreation spaces [44, 86].

Another environmental driver identified was related to the mitigation of the impact of urbanization on the environment through the introduction of changes in urban water management; found in 47% of the cities analyzed (Table 2). These changes were not directly related to the different uses of water but are based on avoiding the degradation of terrestrial and aquatic ecosystems [44]. It is crucial to underscore that the bibliographical review reveals a close correlation between the occurrence of floods and the deterioration of surface water quality, which are both influenced by surface runoff processes intensified by urbanization.

Economic motivations were also found to be a reason for promoting changes in urban water management (Table 2). In 29% of the cities analyzed, the current or potentially future increase in the costs of water management was mentioned as a direct motivation for changing water management policies or actions [44]. For example, in New York City, the city's motivation to implement changes in water management was driven by the potential for significant financial costs associated with the installation of filtration systems, as required by the U.S. Environmental Protection Agency (USEPA) under the United States Safe Drinking Water Act (SDWA), if the city's drinking water quality continued to decline [92]. Likewise, even if it was not reported directly, several of the motivations found in the other cities were closely related to economic issues. For example, the increase in the cost of flood control, the decrease of clean water availability, the interruption of tourism because changes in water quality, the increased cost of cleaning water, among others, were mentioned in the cases even if they were not explicitly stated as drivers of management changes. Even in cities where economic drivers were not the main motivation to start changes, in some cities market incentives facilitated the implementation of some actions. For instance, in Melbourne the construction of third-pipe systems, that is nowadays being installed in many developments around Melbourne's fringes, was promoted by developers since it was considered cost-neutral or actually saving money in comparison with other discharge options [79].

Another identified driver was striving to become more resilient in confronting climate change. However, in the context of the current uncertainty surrounding climate change it is notable that only 24% of the cities analyzed explicitly indicated drivers related to becoming more resilient to face the effects of climate change (Table 2). For example, in Rotterdam, the Netherlands, the primary driver for modifying water management strategies was the necessity to enhance resilience to climate change. Given that Rotterdam is a low-lying port city, particularly vulnerable to rising sea levels as a consequence of climate change, it was imperative to implement measures to mitigate this risk [44, 63].

Lastly, in 24% of the cities, social factors were identified as central motivations for incorporating changes in urban water management (Table 2). For instance, public awareness in water issues and citizens' demand for environmentally friendly policies were drivers for implementing and maintaining a watershed approach in cities such as Berlin, Portland, Melbourne, Singapore, and Vienna [63, 77, 81, 87]. For instance, in Portland, a bottom-up approach was

central for the promotion of a watershed management approach. In this case, citizen advocacy, led by informed and active social stakeholders, filed a lawsuit against the city of Portland, alleging a violation of the Clean Water Act, which triggered Oregon's Department of Environmental Quality (DEQ) to order the reduction of water contamination in the city of Portland [63]. Also, transdisciplinary networks with academics, activists, and corporations, facilitated and promoted a science-based, continuous and incremental implementation of water management changes across different political administrations [72, 79]. Lastly, political factors based on national and local history and geopolitics were key drivers in Berlin and Singapore cases for implementing a watershed approach related to increasing water security and reducing water dependency on imported sources [70–74, 81].

## 3.2 Actions implemented

The identified drivers promoted the incorporation of changes in water management through specific actions in each city (Table 3). The vast majority of cities (76%) implemented actions that represented a shift from gray to green and blue infrastructure. These actions included conversion of impermeable gray spaces in cities to increase infiltration for water availability and flood control using urban design modifications. Actions also included implementation of new infrastructure in cities that considered the water cycle as a priority from project design stages, requiring a strong collaboration between land use planners and water resource managers. Some of the actions implemented were described in the case studies as *Green Infrastructure* (GI), *nature-based solutions* (NBS) *and Low Impact Development strategies* (LID) [69, 71, 74, 92]. For instance, in Berlin, Germany, a series of initiatives were implemented with the objective of sustaining the natural water cycle within the city in order to ensure its water supply remained self-sufficient. Among these initiatives, the filtration of riverbanks, the artificial recharge of groundwater, and the ecological improvement of surface water, as well as the infiltration of rainwater, stand out as particularly noteworthy [45]. Also, in cities of the United States of America, such as Philadelphia and Portland, several actions were taken to promote this shift from gray to green infrastructure. In Philadelphia, storm water regulations and protocols were implemented to increase the use of green infrastructure for storm management by urban developers [91]. Based on those actions, Philadelphia became one of the cities with the highest number of installed green roofs nationally [91]. Similarly, in Portland, based on its "Grey to Green" initiative, over 900 green streets, 400 eco-roofs and 32,000 street trees were implemented to alleviate loadings on the piped infrastructure system, to prevent floods and reduce surface water contamination [63, 90]. Likewise, in Chinese sponge cities the transformation of gray to green infrastructure was central among the implemented actions [62, 66]. For instance, in Wuhan 389 projects were developed including urban gardens, parks and green space for water infiltration [62].

**Table 3. Actions to include a watershed vision in cities' water management based on the selected case studies.**

| ACTIONS | Number of cities implementing this kind of action | Percentage of the total of cities analyzed which implemented this kind of action |
|---|---|---|
| **Shift from gray to green and blue infrastructure** and integration of water as a priority in urban design (nature-based solutions, Low Impact Development strategies, integration of land use planning with water resource management) | 13 | 76 |
| **Rehabilitation or conservation of surface water** and associated ecosystems | 12 | 71 |
| **Alternative water sources** to increase water availability (wastewater reuse, rainwater harvest, or saltwater desalinization) | 7 | 41 |
| **Contamination control policies** | 7 | 41 |

In addition, among the actions identified, the conservation, restoration, and rehabilitation of surface water bodies and their associated ecosystems, such as riparian ecosystems, stand out as particularly noteworthy (Table 3). These types of actions were present in 71% of the cities analyzed and appeared across diverse regions and contexts from Asia to Europe, America, and Oceania. The rehabilitation of bodies of water aims to manage rainwater so that it contributes to flood control. This rehabilitation was also used for runoff control and for natural water treatment to improve water quality through urban wetlands [46]. It is crucial to underscore that these initiatives were implemented through the restoration and rehabilitation of existing bodies of water or through the construction of new bodies of water in close proximity to urban areas. For example, in Melbourne, Australia, the Melbourne Water Living Rivers Program promoted stormwater management devices such as rain gardens and wetlands [79]. Besides, in Singapore, a program began in 2006 to transform its drains, reservoirs and canals into preserved waterways and natural lakes [44]. In cities such as Philadelphia, United States of America, several land acquisitions were implemented (9,200 acres), resulting in the creation of the Fairmount Park, to protect Philadelphia's water sources and, consequently, protecting drinking water quality [91].

Additionally, several actions were implemented to increase water availability from alternative water sources (41% of the cities analyzed) (Table 3). Singapore is an exemplary case in this matter, since it faces severe water availability constraints and implemented a large capital investment to increase water supply. Some of the actions realized by Singapore were increasing the storage and protection of rainwater, generating urban infrastructure for treatment and reuse of wastewater and expanding its desalination capacities [71, 72]. In other cities such as Melbourne, alternative water sources were collected and reused in greenfield areas, such as recycled wastewater for non-drinking uses, rainwater and stormwater harvesting from roofs and after touching the ground [79].

Finally, regarding pollution control policy actions, this type of action was reported in 41% of the cities analyzed (Table 3). For example, in Philadelphia, actions were taken to control combined sewer overflows, which sought to eliminate between 80% and 90% of annual pollutants in combined sewer areas through the use of green infrastructure to control stormwater runoff, even on site [45]. Similarly, pilot projects were designed and implemented in Melbourne based on networks of scientific institutions, policy-makers, land developers, municipalities, and water practitioners. The objective was to increase the use of stormwater quality treatment technologies, such as gross pollutant traps, constructed wetlands, and biofilters [76].

## 3.3 Challenges for the implementation

Despite the drivers of several cities around the world to include a watershed approach in their policies and actions, achieving changes in urban water management towards a watershed approach faces numerous challenges (Table 4).

In several of the urban areas studied, economic investment and market constraints have been identified as significant obstacles to the implementation of watershed-based water management strategies (47% of the cities) (Table 4). The costs of infrastructure building, operation, and maintenance was mentioned as a key challenge to overcome, not only to implement urban water management changes, but also for maintaining, adjusting, and expanding them over time [59, 69]. For example, in Singapore it was found that a barrier to these changes was the high installation and maintenance costs required for new urban water management systems [45]. It is noteworthy that despite the absence of economic investment as a challenge in the Chinese cities included in the case studies, China as a country posits that while it may be economically viable to implement changes in water management in specific cities, economic

**Table 4. Challenges for including a watershed vision in cities water management based on the selected case studies.**

| CHALLENGES | Number of cities facing this challenge | Percentage of the total of cities analyzed which face this challenge |
|---|---|---|
| **Economic investment** (costs of infrastructure building, operation, and maintenance), **and market barriers** (lack of materials needed) | 8 | 47 |
| **Practical and operational barriers** (lack of experience and knowledge of governmental and private implementors, capacities or incentives of stakeholders implementing the actions, or technical and logistical challenges) | 7 | 41 |
| **Social barriers** (social or corporative opposition, lack of interest, engagement, or appropriation by key stakeholders, competing stakeholder interests, or intricated traditional practices and visions around water use and fears around reuse) | 6 | 35 |
| **Institutional and political barriers** (fragmentation, coordination problems, unclear roles, and responsibilities especially for maintenance, water and environmental policies ambiguity, bureaucratic obstacles, or accountability issues) | 5 | 29 |
| **Barriers related to urbanization impacts** in land use generating change from green to gray | 4 | 24 |
| **Barriers for policy implementation by existing legislation** (contradictions with the watershed approach, ambiguities and lagoons in regulations, regulatory barriers associated with human exposure to reused "non-potable" water) | 3 | 18 |
| **Barriers related to geographic characteristics** of the city that make the implementation of some measures more difficult | 2 | 12 |

investment represents a limitation for the implementation of these projects at a basin-wide level [56]. Besides, market barriers were identified, such as local unavailability and high costs of products and materials required for changes in water management, such as Green Infrastructure. The absence of materials locally, such as geo-membranes used in green roofs and rain gardens, increased construction costs and limited a broader implementation of changes in cities such as Seoul [69].

The review also exposed a lack of experience, capacities, and incentives, along with technical and logistical barriers for implementing urban water management changes in governmental and private stakeholders. In 41% of the cities studied, these variables were reported as an obstacle for effectively implementing the watershed approach in their policies and actions (Table 4). For instance, in Melbourne, insufficient knowledge and operational capacities of water retail companies limited their actions for delivering recycled water systems [76]. Additionally, the lack of clarity regarding the determination of necessary treatments within the watershed constituted a barrier in this case [79]. Similarly, the lack of knowledge and experience among city planners in Seoul presented challenges for the effective implementation of LID and GI actions [69].

Additionally, social factors hindered including a watershed approach in urban water management. In 35% of the cities analyzed, the lack of social support was an obstacle for including the watershed approach in water management (Table 4). This was illustrated by the lack of environmental education of the population, the social perception toward new projects and the

need and difficulty of forming interdisciplinary groups to address the issue. In Ningbo, China, even if in the early stages of the planning and construction process for a new water management system, public participation was not a mandatory requirement. The lack of education and communication with stakeholders and the general population impeded future participation in infrastructure maintenance. This was because the beneficiaries should appreciate the social and environmental benefits of the new water management mode, so that the potentially high building and maintenance costs of the model are offset. These cases highlighted the need for appropriation, empowerment, societal support, and economic resources to perform the projects. Therefore, it is suggested that the implementation of new water management models could use the population's participation, information, and consultation, as well as the need for collaboration with the academic sector in order to sustain urban water changes, particularly over reuse of water, on scientific data [64]. A similar situation occurred in Melbourne, where a challenge to implement changes that would facilitate water reuse was the negative opinion and lack of confidence of the population and the political sector regarding the treatment of wastewater to make it available for consumption [79]. Even if the government of Melbourne was convinced of the merits of promoting stormwater harvesting as an alternative water source approach, it was necessary to await the emergence of scientific advancements and new evidence related to biofiltration technologies before proof could be provided that treated water could achieve sufficient quality levels. Until that moment, the context enabled to implement harvested stormwater and recycled wastewater initiatives [76]. Collaborative research and learning process were required to develop evidence, guidelines, and regulations for safely delivering reused water services and to progressively instill trust in political and social actors. It is necessary to emphasize that the potential uses of the treated water depend not only on scientific evidence, but also on community perceptions in each country and locality, for instance, about drinking treated water [76, 79].

Institutional design problems and coordination barriers were also identified as obstacles for implementing the new urban water management in 29% of the cities analyzed (Table 4). The main institutional and political barriers identified were: institutional fragmentation between different areas of the city government; coordination problems between local and national authorities; unclear requirements, roles and responsibilities for implementing and giving maintenance to the new infrastructure generated; and bureaucratic obstacles which problematized the implementation of structural changes [76, 82, 89]. For example, in Melbourne, the difficulty in establishing alliances to employ a new kind of water management was an identified challenge [44]. Additionally, in this city, there was a lack of clear financial evaluation criteria for new water management model projects due to the absence of a systematic approach to quantify the social and environmental benefits of such initiatives, which would justify subsidies and cost-sharing arrangements [79]. In contrast, cities such as Shenzhen where all water-related government functions were combined in a single governmental agency, an integrated management of water permitted to improve water management with a more holistic approach and to avoid conflicts due to overlapping functions and ambiguous functions [65].

The gray infrastructure built in cities and the consequent transformation, fragmentation and degradation of urban green spaces increased the difficulty for introducing changes with a watershed approach in urban water management. For example, 24% of the cities analyzed reported a historical transformation of green spaces into gray spaces as a challenge for including a watershed approach in water management (Table 4). The authors argued that it was easier to include changes in new urban developments than to try to modify the structure of existing urban areas. For example, in Ningbo, in China, converting the old blue and green infrastructure of the city into gray infrastructure was one of the main barriers identified for implementing the Chinese "Sponge City" program [64]. For this reason, the modernization of

the old part of the city was considered harder than the installation of green or blue infrastructure in newly developed areas.

National and local regulations were identified as a barrier for implementing a watershed vision in water management in 18% of the cities analyzed (Table 4). In some cases, the current legislation in their territory discouraged implementing innovative changes in urban water management from a watershed approach. For example, in the case of Berlin, there was a limitation for applying new solutions from research and a pilot project because of the characteristics of the existing legislation [45]. In Melbourne, in highly valued environmental areas where additional stormwater management should occur, it was impossible to impose additional requirements on developers because of the existing regulatory framework [79]. Therefore, existing regulations might be difficult to overcome when applying innovative solutions, both in research and pilot projects [45]. Conversely, in some cities regulation at the national level enabled or even promoted implementing watershed-focused policies. For instance, such as in Chinese cities, changes were guided by national guidelines and specific implementation rules of a national watershed approach [56, 57]. These regulations, national policies, and guidelines were aligned with the change towards watershed-focused actions in water management, facilitating the implementation of these actions at the urban level.

The geographical, geological, and topographic characteristics of the cities were also identified in some cases as an environmental challenge for including a hydrographic watershed approach in urban water management (Table 4). For instance, in two of the cities analyzed, its geographical characteristics were considered a challenge to include water management changes. This was the case of the Chinese city of Chongqing, which is located in a mountainous environment that inhibits engineering solutions for managing high-speed surface runoff from mountains [57]. Also, in the case of Shenzhen, the city's geographical and natural conditions, such as short river channels with limited self-purification capacity, were identified as factors that limits water pollution reduction [65].

## 3.4. Results of the actions

Based on the conducted literature review, it was difficult to extract the impact from the implementation of actions and policies of water management that included a watershed approach (Table 5). The reviewed literature described the actions undertook, their drivers, and challenges, but it rarely evaluated or illustrated the effects of the laws, the policies, and the actions performed by cities to tackle water challenges to move towards sustainability approaches. For some cases, such as the Chinese pilot cities implementing a Sponge City approach, the authors described the cases as having an excellent performance, but they did not specify impact

**Table 5. Results of actions aiming to include a watershed vision in cities' water management based on the selected case studies.**

| RESULTS | Number of cities reporting this result | Percentage of the total of cities analyzed which report this result |
|---|---|---|
| **Increased water availability** (stormwater harvesting, wastewater reuse, groundwater recharge and potable water savings) | 7 | 41 |
| **Ecosystem restoration** | 6 | 35 |
| **Water quality safeguarding** | 6 | 35 |
| **Reduced floods** (Sponge City and Green Infrastructure application) | 5 | 29 |
| **Enhanced habitability in cities and other social benefits** (health, security, recreation) | 3 | 18 |

indicators in terms of sustainability and water issues [57]. Even in Shenzhen and Wuhan cases where the implementation of Sponge City's actions was described as outstanding, the monitoring data was considered insufficient by the authors and basin level ecosystem impacts have not been evaluated [62, 66].

Additionally, some other cases were found in the literature review with a brief description of results from the implementation of water management changes and actions related to a watershed approach incorporation. The literature regarding cities such as Melbourne, Stockholm, and Singapore illustrated a positive impact, which resulted from implementing green infrastructure for rainwater harvesting, drainage networks' interventions, recycled wastewater use, runoff, and floods control. For instance, the Hornsgatan project in Stockholm, which is a rainwater harvesting project based on green infrastructure, was perceived as successful by the community. Some of the emphasized positive results from this project were direct, such as adequate wastewater treatment and water collection from runoff and rooftops; other indirect social benefits included improved air quality, better safety perceived and improved open areas from planted trees that were irrigated mainly through rainwater infiltration [89]. In Melbourne, the incorporation of "integrated water management" provisions in their planning instruments related to rainwater harvesting from roofs and stormwater harvesting (from the ground) was also described as having positive results. For example, the use of stormwater harvesting schemes became an extended practice in Melbourne's urban developments and parks, where citizens reduced the need of purchasing drinking water [79]. In Singapore, rainwater collection and the drainage network in place with grass swales and filter strips, reduced water-logging and flooding throughout the islands even during heavy storms [57]. Collected rainwater was also a new unconventional water supply source there, accounting for 20% of the total water supply sources [45]. In Chongqing's case, an improvement of environmental conditions was identified and water safety was evaluated as having an upward trend [60].

Lastly, it was remarkable that Vienna explicitly presented specific indicators to describe the impact from their actions and water management policies, which included a watershed approach. In this case, the city government reportedly managed to maintain the desired water quality of a surface water body related to tourist use [86]. Moreover, surrounding this water body, a substitution and reforestation of the riparian ecosystem with native species were achieved through public and private cooperation [86].

## 4. Discussion

The literature review allowed the identification of cities that have implemented specific actions with the objective of fostering changes in urban water management. These actions were designed to incorporate hydrological dynamics into the planning, policies, regulations, and projects employed to address water-related issues. For each city, drivers and challenges for implementing those actions, as well as the main results, were analyzed. However, beyond the specific results obtained from this study, it is relevant to discuss the main reflections that emerge from a broader analysis as well as the limitations and gaps of information identified.

### 4.1. Drivers or challenges for urban water management transformation

Whereas some characteristics were identified as drivers for change in some cities, in other cities the same factors were considered as challenges which limited the possibilities of implementing changes in water management. Three factors were identified that shared this duality: a) legislation and policies; b) economic factors; and c) social and political factors.

**Legislation and policies.** The main drivers identified in the cases studied as motivating changes that led to the incorporation of a watershed approach in urban water management

were related to government actions, such as regulations, policies and programs at the national or regional level. The case studies revealed that legislation and policies play a key role in the incorporation of a watershed approach to urban water management. For example, legislation and policies currently in place at the city, state, or country level—as in China and Australia—or even at the regional level, as in the European cases, can be considered key factors. This is because the existing legislation and policies in a given location can be the primary driver for integrating changes in urban water management. A review of the literature revealed the integration of several frameworks that facilitated the incorporation of changes in traditional urban water management as a transition to more sustainable states. Some of these frameworks included the following: low impact development (LID), best management practices (BMPs), sustainable urban drainage systems (SUDS), green infrastructure (GI), the Sponge City (SP), and water sensitive urban design (WSUD) [57, 107].

In addition to the legal and regulatory frameworks that supported these initiatives, in some cases the political and institutional structure also played an important role in facilitating these actions. For example, in the Chinese and Singaporean cases, the governmental structure and coordinated political dynamics allowed for the implementation of changes in water management and urban design, such as the Sponge City program that was implemented in Chinese pilot cities [45, 56, 57]. In other cases, the establishment of rules, processes, and structures to facilitate the alignment and coordination of all aspects of the water cycle in water management and urban planning has been a critical element in implementing changes to incorporate a watershed approach to water management. This required strong collaboration between land use planners and water resource managers to overcome the challenges associated with institutional fragmentation, coordination problems, and ambiguity in roles and responsibilities for implementing changes [76, 82, 89].

Nevertheless, in some cities, the barriers related to legislation, institutions, and their coordination have been effectively overcome. In other cases, however, regulations or institutional structures continued to impede deeper implementation of a watershed approach. It is therefore important to understand, at the city level and in the broader national and regional context, how the local, national and regional legal, policy and institutional frameworks may facilitate or impede the incorporation of changes in urban water management.

Moreover, even in cases where legislation has been aligned with a watershed approach, such as in New Zealand, where a robust legislative framework exists and territorial organization has been based on river basins since 1868, translating frameworks and regulations into action remains a significant challenge [8]. The studies identified in the literature review emphasize that while New Zealand has strong legal and institutional foundations that theoretically enable a watershed approach, other cities, such as those analyzed in this research, have achieved more effective implementation of this approach than New Zealand cities [44, 46].

Additionally, based on the cases and results obtained from the literature review, the incorporation of the watershed approach was identified as primarily driven by the initiative of the local or national government. However, in several countries in the Global South this is not generally the case. In these contexts, due to government deficits, the incorporation of changes in water management frequently occurs as a result of social initiatives and not from governmental initiatives [5]. For example, in Mexico, social initiatives have emerged in contexts with institutional fragmentation, a lack of alignment between the different levels of government (federal, state, and municipal), and in some cases an absence of political will to implement changes [5]. Consequently, the absence of case studies of cities from Global South regions in this review may be due to the complexity of urban systems, which require high levels of investment, political will and coordination from government institutions that may not be present in some Global South contexts.

**Economic factors.**   The fact that all the cities identified in this review belong to countries in the Global North is important because in most of the cases studied, economic investment is a key driver for incorporating watershed approaches in cities. However, even in countries with strong public finances, the investment required to bring about changes in urban water management and to move from pilot cases to a more general and systematic implementation of measures can become a challenge in the long term. Given the economic situation of most countries of the Global South, and the limitations for public economic investment, this factor might represent a strong challenge for the urban transformation of water management towards a watershed approach [102]. Limited government resources and public investment in the Global South may hamper the central role of government in promoting these initiatives, which, according to the case studies, require a high level of government spending to promote the implementation of innovation policies or to create incentives for the private sector to implement these changes. For instance, in an African case study identified in Ethiopia, but excluded by methodological criteria, the lack of governmental political and financial support for incorporating initiatives such as rainwater harvesting methods, was one of the key challenges for amplifying the scope and adoption of these initiatives at the local level [102].

**Social and political factors.**   Even though the implemented actions identified in this review were mainly government initiatives, social and political aspects were also identified in several cases as relevant factors in facilitating, encouraging or at least enabling the implementation of changes. For instance, the public support for the implementation of water management changes that took long time and large resources was fundamental to give some of the initiatives the viability required for being implemented [76, 81, 86]. In contrast, in other cases, social factors were identified as challenges for promoting alternative projects in urban water management. These challenges were identified, for example, in projects focused on the treatment and reuse of wastewater safely [70]. These projects had to face distrust and poor social and political perceptions about this alternative source of water, and actions had to be taken by governments, scientists and social actors promoting these projects to change social perceptions and overcome this barrier. For example, Singapore's strategies for public acceptance of wastewater and rainwater reuse could be a relevant lesson for other cities seeking to promote these actions. Singapore has developed various education and communication strategies to raise public awareness of water issues and change attitudes towards wastewater, even replacing "wastewater" with "used water" in public language to reinforce the idea that this is an important resource that needs to be recycled and reused to ensure urban water sustainability and climate change resilience [70]. This is an example of a city transforming a social barrier into a social driver, by creating a change in the social context that allowed for the effective implementation of new water management initiatives.

## 4.2. Actions to integrate urban watersheds from a socio-ecological perspective

Two general categories of actions were identified among the cities studied. On the one hand, a group of actions focused on reversing the impacts of urbanization processes, through the restoration of rivers or urban green spaces. These actions represent most of the actions identified in the case studies and seek to recover the benefits of ecosystems by preserving or restoring natural processes. On the other hand, there are actions that seek to incorporate the understanding of variables, dynamics and natural processes through innovations such as rainwater harvesting or reuse of treated water.

A crucial element identified in the first group of actions, is the effect of land use change promoted by urbanization. The degree of fragmentation, reduction, and disappearance of natural

spaces (urban green spaces and surface water bodies), as well as their degree of conservation must be considered in order to incorporate a watershed approach to urban water management. These characteristics of urban land use will hinder or promote the incorporation of changes in water management from a watershed approach and will determine the level of complexity and socio-economic costs that the proposed changes in urban water management would imply. For instance, 71% of the cities analyzed addressed actions for the conservation, restoration or rehabilitation of surface water bodies and their associated ecosystems, which implies a high degree of economic investment and other challenges to overcome in order to recover natural processes or dynamics that benefit the city. This investment makes it difficult and very costly to implement these types of actions, and it is particularly in cities of the Global South. In this sense, cities that maintain a higher level of natural spaces conservation could achieve a more solid incorporation of a watershed approach and face fewer challenges with lower implementation costs. This highlights the role of natural spaces in cities and the importance of territorial and ecological planning that promotes the conservation of natural water processes and dynamics in the urban areas.

A second group of actions identified through the case studies are the innovations implemented, such as rainwater harvesting and the reuse of treated water. These types of innovations aim to integrate water dynamics into urban planning and to adapt the technology of use to the hydrological dynamics. Regarding these innovations, something that was rarely present in the literature review, but which is undoubtedly essential to consider when thinking about this type of innovation, is the relationship of the cities with climate change projections and the uncertainty that this represents [108, 109]. Climate change scenarios add complementary difficulties and uncertainties for water management that must be considered with a view of more sustainable and resilient cities in a short term. This is evidenced by the fact that changes in urban water regimes pose a growing risk and have already led to the collapse of water supply, stormwater and wastewater disposal systems in several cities around the world. Even in cities with a strong tradition of a watershed approach, such as Melbourne, technical analyses for the implementation of water management measures do not take into account future climate change scenarios, for example in the case of rainwater harvesting systems, which are climate-dependent [78].

In both types of actions, either ecosystem restoration actions or the promoted innovations, the incorporation of a watershed approach in urban water management is facilitated if it is based on understanding and respecting the physical and biophysical components of the system of the city and its watershed [110, 111]. This promotes the existence of resilient ecosystems (biophysical components) in which cities and their inhabitants can live and sustain ecological services as well as their common goods. Thus, a socio-ecosystem approach is useful to achieve because it understands water systems as composed of physical components in which the biophysical as well as human, social and institutional components are nested [111]. This type of alignment promotes long-term benefits of blue and green spaces, such as those identified in the case studies, which outweigh short-term economic savings and reduce future investments to restore lost blue and green spaces.

**4.3. The limitation of the information about the results.** It is important to highlight that it was difficult to extract from the literature review conducted the results of the implementation of water management actions and policies that include a watershed approach. The literature reviewed described the actions taken, their drivers and challenges, but rarely evaluated or illustrated the impact of laws, policies and actions taken by cities to address water challenges and to move towards sustainability approaches. We could relate this to the fact that, as mentioned above, the studies found belong to recent years and therefore, as we are dealing with natural processes and dynamics, time is needed to observe the results. However, no indicators

have been identified to evaluate these results and to promote an adaptive management, which is necessary for ecosystem management [112]. Therefore, a vacuum and an opportunity have been observed, based on the review carried out, to strengthen the scientific studies and policy instruments related to the monitoring and evaluation of the impact of the actions taken into the problems that cities are willing to solve.

**4.4. Major limitations and gaps of this study.** The analysis of the literature reviewed allowed us to identify that all the papers reviewed were from the last 25 years and the majority were from the last decade. This shows that, although urban water management and the problems associated with it are a traditionally subject, the incorporation of a watershed approach to actions taken in the urban area is a contemporary issue. Interestingly, based on the methodology and specific keywords used, all the cases identified are cities in countries of the Global North or in emerging world powers, such as Singapore, South Korea, and China.

Some of the factors that may influence the lack of case studies from cities in the Global South as a result of this research are the lack of intergovernmental coordination in incorporating a watershed approach, as civil society actors have a stronger role in some contexts. In addition, the need for strong public financial support for introducing some of the changes in water management and implementing innovations in cities related to incorporating a watershed approach in urban areas may also explain the lack of case studies from the Global South in this research. In addition, many of the papers identified in this scoping review in Africa, Asia and Latin America take place in rural areas or are conceptual proposals and scenarios for possible changes in urban water management that aim to promote more sustainable water management policies [101, 105, 113]. Based on a methodological decision and the inclusion and exclusion criteria applied, this type of studies was not analyzed in this review, as they were not actions carried out in urban contexts. Thus, for future research, it would be necessary to analyze what actions are being implemented in the Global South regarding urban water management, the role governmental institutions and social actors play, and what are their main drivers, actions, challenges, and results.

Based on the characteristics of this review, which included only actions implemented in cities, this scoping review has an operational limitation that cannot reflect the great heterogeneity of cities worldwide and the initiatives implemented by several countries that are not specifically located in a city, but have instead a more regional or national focus of implementation. For example, some cases from Canada were excluded from this review because they were not specifically located in a city and described regional actions at the watershed level across several interconnected regions [95, 114]. It is necessary to recognize the heterogeneity of urban areas, with semi-urban and peri-urban areas, and the relevance of urban-rural interactions not only at the urban level, but also at the regional and national levels in order to move towards more sustainable states of urban water management.

Finally, the databases reviewed, the language and the number of documents reviewed in each database were also limited due to the time and resources available. In addition, the documents reviewed in this scoping study are current until January 2024. Therefore, documents published and appearing in databases after this date were not included in this scoping review. Due to the above characteristics and limitations of the scoping review, some cities' cases may have been missed. In particular, the authors note a limitation in the lack of case studies from arid regions where water scarcity is a major concern, but which, due to the methodology used, were not identified as a result of this research. Therefore, further specific analysis of these cases in complementary research would be necessary to take into account the specific characteristics, actions and challenges related to urban water management in these contexts, where initiatives such as green and blue space preservation and rainwater harvesting are less relevant and innovations such as water reuse may become more relevant [115].

## 5. Conclusion

This research provides an analysis of specific cases of cities that have implemented actions to incorporate a watershed approach to urban water management. This study identifies the drivers, the actions, the challenges and the results obtained from the application of these actions worldwide. This differs from other existing studies and literature reviews related to the watershed approach and to water management. Traditionally, studies focus on conceptual models of watershed management, analyze single case studies, regional or basin scale cases, cases from rural cases and natural areas from non-urban locations. This study has an innovative focus as it generates valuable information on what has been done in cities worldwide to actually incorporate, with specific actions, and within urban contexts, hydrological dynamics that tend to be neglected and therefore excluded from water management and urbanization processes, decisions, and actions.

Through this research we were able to identify how the incorporation of natural dynamics and processes related to water has been translated into specific actions for urban water management. This is an issue that has been presented as a historical priority and scholars have consistently highlighted the challenges of actually implementing a watershed approach. Because of these challenges, cities have had to deal with the same problems of water scarcity, flooding, sewage, and water contamination for many years, and these continue to be the main drivers for generating changes in water management policies and incorporating a watershed approach. The technological solutions that have helped to address these problems have increased economic, social, and environmental costs, and no longer seem to be the most effective and efficient solutions. Moreover, in the face of climate change, water issues are becoming even more urgent to address from a sustainable perspective.

Based on the results of this study, the relevance of green and blue spaces in cities was highlighted for the effective implementation of a water approach to water management. Furthermore, the need to align national and local regulations and policies was also identified as a key element to promote the actual and effective implementation of projects aimed at generating changes in urban water management from a watershed perspective.

Urban water management, instead of being a promoter of negative urban and regional impacts, could be seen as an opportunity to change the way we relate to the urban territory, by understanding that the positive and negative impacts of cities have reaches beyond the boundaries of their territory, but rather they have a regional and global impact. Therefore, incorporating a watershed approach to urban water management could promote a more comprehensive management of the territory through inclusive watershed management that can promote more sustainable conditions at the local, regional and global levels.

The methodological proposal of this research could be used as a basis for future studies in order to analyze complementary cases that were not included in this research. Based on the same methodology, it could be possible to compare the results obtained in this study with other scenarios with a variety of social, political, economic and environmental contexts. This iterative and comparative analysis could be valuable in promoting a more effective and feasible implementation of the watershed approach and seems to be key in moving towards more sustainable states.

Finally, the results presented in this research highlight an opportunity, particularly for small and medium-sized cities that preserve green and blue spaces in the Global South, to adopt a watershed approach. These cities may be able to implement a watershed approach on a more efficient and cost-effective basis by preserving the natural areas that already exist in their territories and by preventing urban land changes in areas that are critical for water dynamics. If considered by authorities at this stage of their urban growth, this could reduce the cost of

incorporating a watershed approach, in contrast to more urbanized cities that have already highly severely degraded or urbanized these natural spaces. This opportunity is important to move towards a watershed approach to water management that can lead to more sustainable future for these cities and their surrounding areas.

## Supporting information

**S1 File. PRISMA-ScR—checklist.**
(DOCX)

**S2 File. Descriptive summary of the documents included in this scoping review.**
(DOCX)

**S3 File. Studies identified in the literature search.**
(DOCX)

## Acknowledgments

Marcelo Canteiro has a postdoctoral position and would like to thank the "*Programa de Becas Posdoctorales de la Dirección General de Asuntos del Personal Académico*" of the *Universidad Nacional Autónoma de México* (UNAM). Nadjeli Babinet is a PhD student and would like to acknowledge the "*Posgrado en Ciencias de la Sostenibilidad*" of the UNAM. The authors thank Raúl Ahedo Hernández, Heberto Ferreira Medina, Alberto Valencia García and Atzimba López Maldonado for their technical support.

## Author Contributions

**Conceptualization:** Marcelo Canteiro, Helena Cotler, Marisa Mazari-Hiriart, Nadjeli Babinet, Manuel Maass.

**Data curation:** Marcelo Canteiro, Nadjeli Babinet.

**Formal analysis:** Marcelo Canteiro, Nadjeli Babinet.

**Investigation:** Marcelo Canteiro.

**Methodology:** Marcelo Canteiro, Nadjeli Babinet, Manuel Maass.

**Resources:** Manuel Maass.

**Supervision:** Manuel Maass.

**Validation:** Helena Cotler, Marisa Mazari-Hiriart, Manuel Maass.

**Writing – original draft:** Marcelo Canteiro, Manuel Maass.

**Writing – review & editing:** Marcelo Canteiro, Helena Cotler, Marisa Mazari-Hiriart, Nadjeli Babinet, Manuel Maass.

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
