## [Decision Letter · Decision Letter 0]

9 May 2024

PONE-D-24-10058Natural dynamics and watershed approach incorporation in urban water management: A scoping reviewPLOS ONE

Dear Dr. Canteiro,

Thank you for submitting your manuscript to PLOS ONE. After careful consideration, we feel that it has merit but does not fully meet PLOS ONE’s publication criteria as it currently stands. Therefore, we invite you to submit a revised version of the manuscript that addresses the points raised during the review process.

The primary focus of this scoping review study has significant value for urban watershed management. However, both reviewers have identified critical methodological gaps; and the need to strengthen the rational, discussion and conclusions of this article. Specifically, both reviewers identified study sampling bias issues that must be addressed. For example, reviewer 1 states: "The paper does recognize the lack of any representation from the Global South. In the discussion section, the authors do explore the consequences of social, economic, and policy challenges that might be encountered in developing-country settings. This discussion, although superficial in nature because it represents a knowledge gap in the study, does not mention the situation in cities in developing countries within Africa and Asia. That is inexplicable gap should be addressed easily." Along the same lines, Reviewer 2 recommends: "The authors extensively discuss the lack of studies in Latin America. But there seems like other gaps as well (e.g. efforts in Canada, Japan). Perhaps this could be explored a bit more – why did the search criteria leave these out and what are the impacts? Was this a methodology limitation?"

Limitations of the scope of this study, and gaps for future research may be more clearly identified in the discussion and/or methodology sections.

All major and minor issues raised by both reviews must be addressed in the revised version.

We look forward to receiving your revised manuscript.

Kind regards,

Asim Zia, Ph.D.

Academic Editor

PLOS ONE

Journal Requirements:

Additional Editor Comments:

The primary focus of this scoping review study has significant value for urban watershed management. However, both reviewers have identified critical methodological gaps; and the need to strengthen the rational, discussion and conclusions of this article. Specifically, both reviewers identified study sampling bias issues that must be addressed. For example, reviewer 1 states: "The paper does recognize the lack of any representation from the Global South. In the discussion section, the authors do explore the consequences of social, economic, and policy challenges that might be encountered in developing-country settings. This discussion, although superficial in nature because it represents a knowledge gap in the study, does not mention the situation in cities in developing countries within Africa and Asia. That is inexplicable gap should be addressed easily." Along the same lines, Reviewer 2 recommends: "The authors extensively discuss the lack of studies in Latin America. But there seems like other gaps as well (e.g. efforts in Canada, Japan). Perhaps this could be explored a bit more – why did the search criteria leave these out and what are the impacts? Was this a methodology limitation?"

Limitations of the scope of this study, and gaps for future research may be more clearly identified in the discussion and/or methodology sections.

All major and minor issues raised by both reviews must be addressed in the revised version.

Reviewers' comments:

Reviewer's Responses to Questions

**Comments to the Author**

1. Is the manuscript technically sound, and do the data support the conclusions?

Reviewer #1: Yes

Reviewer #2: Yes

2. Has the statistical analysis been performed appropriately and rigorously? 

Reviewer #1: Yes

Reviewer #2: N/A

3. Have the authors made all data underlying the findings in their manuscript fully available?

Reviewer #1: Yes

Reviewer #2: Yes

4. Is the manuscript presented in an intelligible fashion and written in standard English?

Reviewer #1: Yes

Reviewer #2: Yes

5. Review Comments to the Author

Reviewer #1: The paper presents a summary review of using watershed management approaches in urban settings to manage water resources. In the 17 case studies analyzed, the authors have identified some common drivers for such approaches being explored by cities (scarcity, flooding, pollution), and also the challenges in successful implementation (lack of funding, inexperience, stakeholder opposition, and regulatory obstacles). The authors argue that a successful urban implementation of such water management approaches can also trickle upwards to larger jurisdictions, e.g. to state-wide or national implementation.

This literature-review based study offers some interesting and useful insights. These insights could form the basis of further dialogues in the cities included in the paper, and others with similar social, economic, and policy situations. In that respect, the paper serves an important function.

The paper suffers from two major biases:

First, the paper does recognize the lack of any representation from the Global South. In the discussion section, the authors do explore the consequences of social, economic, and policy challenges that might be encountered in developing-country settings. This discussion, although superficial in nature because it represents a knowledge gap in the study, does not mention the situation in cities in developing countries within Africa and Asia. That is inexplicable gap should be addressed easily.

Second, all the cities included in the study do not fall in arid regions; this is a gap that the authors did not recognize or acknowledge. For cities in arid climates, water scarcity is typically a much bigger concern (even when located in developed countries – e.g., Las Vegas in the United States), and water management takes on a much higher degree of urgency in terms of water management, policy, and politics. The authors must: (a) recognize this shortcoming of the study explicitly; and (b) identify how this bias impacts their discussion and conclusions.

Here are some specific comments:

- Page 3, line 64. Increasing urban water consumption is also correlated to water losses (non-revenue water) in aging infrastructure (mentioned on line 135). Please identify this as a key factor in water availability, even if the documents you reviewed did not explicitly identify it.

- Page 10, line 214. Please list some of the Chinese cities, just as is done for Australian and American cities.

- Page 12, line 258. Please provide a listing of the research questions, perhaps as an Annex.

Reviewer #2: This paper applies a well described systematic methodology to reviewing existing literature focused on the implementation of watershed management approaches in urban settings. I believe the methodology is generally sound and well described. However, I have some concern that this methodology may have limited the number of relevant studies included in the review (and this is acknowledged by the authors). More importantly, as there are a number of existing publications, including review papers, that focus on watershed management approaches, I think some additional thought and discussion of what makes this review different would be valuable to readers. The answer to the question “What is this paper offering that others have not?” should be immediately identifiable. In addition, I believe some further editing work would improve readability and make the main points of this review more impactful.

6. PLOS authors have the option to publish the peer review history of their article (what does this mean?). If published, this will include your full peer review and any attached files.

Reviewer #1: **Yes: **Zafar Adeel

Reviewer #2: No

---

## [Author Response · Author response to Decision Letter 0]

27 Jun 2024

We are pleased to send you a revised version of the manuscript "Natural dynamics and watershed approach incorporation in urban water management: A scoping review". We considered all the comments and appreciate the suggestions made by the reviewers. We are convinced that their comments and recommendations helped to significantly improve the manuscript. Changes were addressed throughout the manuscript and the detailed answers are presented below. Additionally, a "Revised Article with Changes Highlighted” is included where all changes are marked through the “Track Changes” option in Microsoft Word.

Reviewer #1:

1: First, the paper does recognize the lack of any representation from the Global South. In the discussion section, the authors do explore the consequences of social, economic, and policy challenges that might be encountered in developing-country settings. This discussion, although superficial in nature because it represents a knowledge gap in the study, does not mention the situation in cities in developing countries within Africa and Asia. That is inexplicable gap should be addressed easily.

2: Second, all the cities included in the study do not fall in arid regions; this is a gap that the authors did not recognize or acknowledge. For cities in arid climates, water scarcity is typically a much bigger concern (even when located in developed countries – e.g., Las Vegas in the United States), and water management takes on a much higher degree of urgency in terms of water management, policy, and politics. The authors must: (a) recognize this shortcoming of the study explicitly; and (b) identify how this bias impacts their discussion and conclusions.

R1-2: Thank you for these comments, since it helped to improve the discussion of the article and to avoid bias in the interpretation of our results and for identifying the gaps of the study. To address these comments, modifications were made to the entire discussion section in order to mention the gaps encountered in different regions of the Global South, including Africa and Asia, as well as highlighting the lack of cases from arid regions and to recognize how this can generate a bias in the results exposed. This was incorporated in the section “Main limitations and gaps of this study” of the Discussion, between lines 944 to 994.

3: Page 3, line 64. Increasing urban water consumption is also correlated to water losses (non-revenue water) in aging infrastructure (mentioned on line 135). Please identify this as a key factor in water availability, even if the documents you reviewed did not explicitly identify it.

R3: The sentence was rewritten to include water leaks in aging infrastructure as a key factor in water availability. Changes are marked in the text between lines 74 to 80 on page 4.

4: Page 10, line 214. Please list some of the Chinese cities, just as is done for Australian and American cities.

R4: Added examples of some of the Chinese cities identified in the review on lines 210 and 211.

5: Page 12, line 258. Please provide a listing of the research questions, perhaps as an Annex.

R5: The research question that motivated this scoping review is explicitly stated at the end of the introduction section between lines 233 to 234. In addition, the "Step 1: Research questions" section of the methodology was improved and the list of specific research questions was added based on the reviewer's comment. The changes are marked in the text between lines 252 to 262.

Reviewer #2:

1: I have some concern that this methodology may have limited the number of relevant studies included in the review (and this is acknowledged by the authors). More importantly, as there are a number of existing publications, including review papers, that focus on watershed management approaches, I think some additional thought and discussion of what makes this review different would be valuable to readers. The answer to the question “What is this paper offering that others have not?” should be immediately identifiable. In addition, I believe some further editing work would improve readability and make the main points of this review more impactful. 

R1: This comment allowed us to strongly improve the article. Based on this suggestion, specific changes were made throughout the entire document, but fundamentally in the introduction and the conclusion sections to highlight the importance and the novelty of the work, answering more clearly the reviewer’s question about what this paper offers that other have not, and to make explicit the difference with other studies. Modifications were made to the document between lines 218-231 in the introduction section, as well as in the conclusion section (lines 996-1007).

2: General editing, while I have provided a few examples and suggested revisions here, I would suggest a thorough editing of the entire paper to ensure a consistent writing style, consistent use of tense, improve succinctness and reduce redundancy, and improve clarity of sentences and paragraphs. Each paragraph should have a primary idea/message that it communicates to readers. Below are some examples:

R2: The entire text was reviewed and edited, errors identified were corrected, and repetitions were eliminated, in order to improve the writing and make it clearer for readers. The reviewer examples helped to improve the clarity of several sentences in the text. Regarding the specific phrases proposed by the reviewer to improve the text, they were all incorporated and the lines where these modifications were made are indicated below. Likewise, phrases with similar problems throughout the text were located and improved.

3: Lines 50-51 (particularly important example b/c lead paragraph/sentence): “because of their need to obtain water to supply their population, their commercial and productive activities, and their industry, as well as facing the scarcity of this resource” -> “because of the conflicting demands on and availability of this resource. While urban centers have high needs due to their higher population densities as well as commercial and industrial activities, they often have higher scarcity of this resource due to {maybe add reason here}”. 

R3: Based on the reviewer's suggestion, the wording of this sentence was modified to improve its clarity; for this purpose, modifications were made between lines 47 to 51.

4: From abstract and lines 23-25: “Although the importance of a watershed approach in land management is acknowledged, when studies focus on water management, they generally study cases in rural areas or single case-studies of cities or countries” -> “Although the importance of a watershed approach in land management is generally acknowledged, studies on this topic have typically focused on water management in rural areas or single case-studies of cities/countries.” 

R4: We understand that the reviewer's suggestion improves the clarity of the sentence, therefore modifications were made based on said suggestion between lines 22 to 25.

5: Lines 21-22: “Some cities, however, have intended to incorporate a watershed approach in water management to seek more sustainable solutions.” -> “Some cities have incorporated a watershed approach to water management in seeking more sustainable solutions.”

R5: The sentence was rewritten based on the reviewer's suggestion to improve it clarity between lines 21- 22.

6: Lines 70 -71: “This risks urban water being contaminated by microbiological agents and organic and inorganic compounds…” -> “The risks include urban water being …”

R6: The sentence was rewritten based on the reviewer's suggestion between lines 58 - 59.

7: New paragraph at line 92 with focus on transport/regional effects of urbanization? There are a few sentences on the effects of urbanization sprinkled through the introduction – maybe consolidate for more impact?

R7: Based on this comment, the urbanization impacts section was improved, consolidating in a single paragraph the ideas to provide greater clarity and impact of the desired message. In order to achieve this goal, the section was separated into two parts, one with the impacts within the urban area and another with the regional impacts. The changes are marked in the text of the document from line 56 to line 100.

8: Lines 391-393 seem a bit contradictory – states that the primary issue is water quantity issues, but then water quality is listed as a motivation behind that. This paragraph could use clarification. I think the overall structure you are going for is a paragraph on each of the items in Table 2? If so, perhaps try to stick with that and focus as tightly as you can within each paragraph.

R8: The wording of this sentence, between lines 404 and 414, was improved to achieve greater understanding of the idea and avoid contradictions.

9: Line 423: “…in Chinese’s cases, …” – should be either “…in China’s cases,…” or “…in Chinese cases,…” 

R9: The error in the expression was solved based on the reviewer's comment on line 441.

10: Line 486-488: “…which triggered that Oregon’s Department of Environmental Quality (DEQ) ordered the reduction of water pollution to the city of Portland ” ->”…which triggered Oregon’s Department of Environmental Quality (DEQ) to order the reduction of water pollution in(?) the city of Portland.”

R10: The sentence was rewritten based on the reviewer's suggestion to improve it clarity between lines 505 - 506.

11: Lines 499-501: “These actions included converting impermeable gray spaces existent in cities, increase infiltration for water availability and floods control, using urban design modifications.” -> Unless I’ve misunderstood, the use of commas here is incorrect -> “These actions included conversion of impermeable gray spaces in cities to increase infiltration for water availability and flood control using urban design modifications.”

R11: The sentence was rewritten based on the reviewer's suggestion to improve it clarity between lines 518 – 520.

12: Lines 501-503: “Also, this type of actions included the implementation of new infrastructure in cities incorporating the water cycle as a priority since project design, requiring a strong collaboration of land use planners with water resource managers.” -> “Actions also included implementation of new infrastructure in cities that considered the water cycle as a priority from project design stages, requiring a strong collaboration between land use planners and water resource managers.”

R12: The sentence was rewritten based on the reviewer's suggestion to improve it clarity between lines 520 - 522. 

13: Review uses of clauses like ‘Thus’, ‘Furthermore’, ‘In order’, ‘With regard to’, ‘Moreover’ -> these are often not necessary and in some cases not used correctly, taking away from the main point of sentences.

R13: The use of this type of expression throughout the text was reviewed and the necessary modifications were made to make the text clearer and to guarantee the correct use of these expressions. 

14: Table 1: add state to other US cities

R14: In the previous version of this article, the name of the State had been only added for the city of Portland (USA) since there are two cities in the country with the same name in different states. However, based on the reviewer's suggestion, all the states, provinces, or regions of the cities used as case studies in this research were added to Table 1 in line 368 on page 17.

15: Table 2, 3, 4, and 5: I’m struggling to understand the % number here – perhaps explicitly state the denominator. It is also unclear whether number of cases indicates number of cities/urban centers where this was the driver, or number of studies more generally? Can multiple drivers have been identified for a single urban center?

R15: In tables 2 to 5, the titles were improved for greater clarity of the data shown. This was done by changing the column title from: “N° of cases” to “Number of cities presenting this…” driver, action, challenge or result, depending on the section, and the column title “%” to “Percentage of the total of cities analyzed which present this...” driver, action, challenge or result, depending on the section. Additionally, a paragraph was added before the tables 2, 3, 4 and 5 are presented in order to give a clearer explanation about the percentages presented in the table and the number of cases. The paragraph incorporated was the following: “Those results have been organized in Tables 2 to 5 which show the main categories of drivers, actions, challenges, and results identified based on the cities analyzed. For each topic, the tables indicate the number of cities that fall under each category (main drivers, actions, challenges and results), as well as the percentage that this number represents over the total cases of cities analyzed.” (Lines 400 to 402) 

16: In the text (primarily Methodology section), does ‘case’ refer to a single city/urban center, or a single study, or are these the same? 

R16: The entire text was reviewed, particularly for the Methodology section, and the necessary modifications were made to avoid confusion regarding the meaning of the expression "cases". The majority of the changes done in order to avoid confusions were realized by converting the word "cases" to "cities", to clarify the focus of this investigation and its results.

17: Discussion: The authors extensively discuss the lack of studies in Latin America. But there seems like other gaps as well (e.g. efforts in Canada, Japan). Perhaps this could be explored a bit more – why did the search criteria leave these out and what are the impacts? Was this a methodology limitation?

18: Discussion: The methodology is described from the standpoint of the drivers, actions, challenges, and results. However, the discussion is not clearly framed in the same way so I find it a bit challenging to tie together the results and discussion directly.

19: Discussion: I think there’s something interesting here related to the lack of evidence on results of implementation, and lack of metrics, that could be brought up in the discussion as well.

R17-18-19: These comments undoubtedly promoted an improvement in the quality and the clarity of the discussion section. The comments were addressed throughout all the discussion section, between lines 757 to 994. This section was reviewed and restructured in order to address all the comments of the reviewer. For instance, the new structure of the discussion is tied to the structure of the results, discussing the interaction between drivers and challenges, then the main reflections related to the actions and then highlighting the gaps in the results identified. Lastly, the limitations and gaps identified are explained, undertaking a deeper discussion of other gaps mentioned by the reviewer, such as efforts in other countries of the Global North (such as Canada) and integrating a deeper explanation of the exclusion criteria and its implications.

20: Conclusions: Highlight here again what this paper offers that others haven’t and then the main take-aways, as supported by the review process implemented here. This should not just be a summary of the paper, it should be the key findings and perhaps what can we build on, etc.

R20: Based on this comment, the introduction and the conclusion sections were modified and improved in order to provide more clarity of the contribution of the work and its difference with other similar works. Modifications were made to the document between lines 218-228 in the introduction section, as well as in lines 996-1007 of the conclusions.

---

## [Decision Letter · Decision Letter 1]

8 Aug 2024

Natural dynamics and watershed approach incorporation in urban water management: A scoping review

PONE-D-24-10058R1

Dear Dr. Canteiro,

We’re pleased to inform you that your manuscript has been judged scientifically suitable for publication and will be formally accepted for publication once it meets all outstanding technical requirements.

Kind regards,

Asim Zia, Ph.D.

Academic Editor

PLOS ONE

Additional Editor Comments (optional):

Reviewers' comments:

Reviewer's Responses to Questions

**Comments to the Author**

1. If the authors have adequately addressed your comments raised in a previous round of review and you feel that this manuscript is now acceptable for publication, you may indicate that here to bypass the “Comments to the Author” section, enter your conflict of interest statement in the “Confidential to Editor” section, and submit your "Accept" recommendation.

Reviewer #2: All comments have been addressed

2. Is the manuscript technically sound, and do the data support the conclusions?

Reviewer #2: Yes

3. Has the statistical analysis been performed appropriately and rigorously? 

Reviewer #2: N/A

4. Have the authors made all data underlying the findings in their manuscript fully available?

Reviewer #2: Yes

5. Is the manuscript presented in an intelligible fashion and written in standard English?

Reviewer #2: Yes

6. Review Comments to the Author

Reviewer #2: I recognize and appreciate the amount of effort you put in to address all comments - I do think it has resulted in a much-improved paper.

7. PLOS authors have the option to publish the peer review history of their article (what does this mean?). If published, this will include your full peer review and any attached files.

Reviewer #2: No

---

## [Editor Report · Acceptance letter]

22 Aug 2024

PONE-D-24-10058R1 

PLOS ONE

Dear Dr. Canteiro, 

I'm pleased to inform you that your manuscript has been deemed suitable for publication in PLOS ONE. Congratulations! Your manuscript is now being handed over to our production team.

Kind regards, 

on behalf of

Professor Asim Zia 

Academic Editor

PLOS ONE